# Off-Team Learning

**Brandon Cui**[†]
MosaicML
brandon@mosaicml.com

**Hengyuan Hu**[†]
Stanford University
hengyuan@cs.stanford.edu

**Andrei Lupu**
Meta AI & FLAIR, University of Oxford
alupu@meta.com

**Samuel Sokota**
Carnegie Mellon University
ssokota@andrew.cmu.edu

**Jakob N. Foerster** [†]
FLAIR, University of Oxford
jakob.foerster@eng.ox.ac.uk

## Abstract

Zero-shot coordination (ZSC) evaluates an algorithm by the performance of a team of agents that were trained independently under that algorithm. *Off-belief learning* (OBL) is a recent method that achieves state-of-the-art results in ZSC in the game Hanabi. However, the implementation of OBL relies on a belief model that experiences covariate shift. Moreover, during ad-hoc coordination, OBL or any other neural policy may experience test-time covariate shift. We present two methods addressing these issues. The first method, off-team belief learning (OT-BL), attempts to improve the accuracy of the belief model of a target policy $\pi_T$ on a broader range of inputs by weighting trajectories approximately according to the distribution induced by a different policy $\pi_b$. The second, off-team off-belief learning (OT-OBL), attempts to compute an OBL equilibrium, where fixed point error is weighted according to the distribution induced by cross-play between the training policy $\pi$ and a different fixed policy $\pi_b$ instead of self-play of $\pi$. We investigate these methods in variants of Hanabi.

## 1   Introduction

A core objective of multi-agent reinforcement learning (MARL) is to develop methods for coordinating with previously unseen partners such as other artificial agents or humans. In partially observable fully cooperative settings, policies that were learned through self play develop efficient but opaque arbitrary conventions that make collaboration with novel partners, or even independent training runs of the same algorithm, extremely difficult [6, 4]. To study this issue, Hu et al. [6] introduced the zero-shot coordination (ZSC) setting in which the goal is to maximize the intra-algorithm cross-play (intra-AXP).[2] ZSC evaluates algorithms under intra-AXP, i.e. by independently training $N$ groups of agents and evaluating teams made up of one member of each group. For an algorithm to achieve good intra-AXP, it must tend to find compatible strategies across different, independent training runs.

Off-belief learning (OBL) [7] is a recent MARL algorithm designed to produce high intra-AXP. OBL takes in an input policy $\pi_0$ and aims to produce a uniquely defined output policy $\pi_1$. It effectively

---

[†]Work done while at Meta AI (FAIR).

[2]Note that the term "zero-shot coordination" is used ambiguously in the original paper[6], both informally and as a specific problem setting. To avoid confusion, in this paper we use it strictly for the technical setting.

36th Conference on Neural Information Processing Systems (NeurIPS 2022).

uses $\pi_0$ to reinterpret past actions as if they were taken by $\pi_0$. When $\pi_0$ is a fully random policy, this forces $\pi_1$ to only take actions based on *grounded* information. This also means that $\pi_1$ has no incentive to communicate private information since any behavior will be reinterpreted as having come from $\pi_0$. OBL then iterates upon this $\pi_1$ by training a hierarchy where each level builds upon the conventions induced by optimal play at the level below. For each $\pi_\ell$, OBL trains a belief model $\hat{\mathcal{B}}_{\pi_\ell}$ through supervised learning, which is then used to interpret past actions when training $\pi_{\ell+1}$.

While OBL achieves strong results in ZSC on Hanabi [1], there are two problems with the current formulation of OBL [7]. First, the belief model $\hat{\mathcal{B}}_{\pi_\ell}$ is *trained* via supervised learning on the distribution of trajectories induced by self-play between $\pi_\ell$. However, it is *evaluated* on the trajectories induced by self-play with the OBL policy at the next level, $\pi_{\ell+1}$. When the distributions of $\pi_\ell$ and $\pi_{\ell+1}$ differ substantially, this will bring $\hat{\mathcal{B}}_{\pi_\ell}$ off-distribution, which may cause it to be unreliable and produce undefined outcomes. Second, OBL policies may be brittle when paired with a novel test time partner, e.g. a human or another AI agent. Note that the second problem is prevalent in most neural policies trained with RL against specific partners.

To mitigate the covariate shift experienced by $\hat{\mathcal{B}}_{\pi_\ell}$ we propose off-team belief learning (OT-BL). Rather than training $\hat{\mathcal{B}}_{\pi_\ell}$ only on the distribution induced by $\pi_\ell$, we introduce a belief-bootstrapping operator that allows us to train our belief model *off team* on the distribution partly induced by $\pi_{\ell+1}$. We evaluate OT-BL empirically and show that it ameliorates the covariate shift experienced by belief models when training OBL and outperforms vanilla OBL in a 2-life variant of the card game Hanabi.

Next, to mitigate the test time covariate shift experienced by $\pi_\ell$ when paired with an unseen agent, we propose off-team off-belief learning (OT-OBL). OT-OBL builds on the insight that OBL can be applied to *off team*. In another word, thanks to the reinterpretation idea of OBL, we can not only train on data generated from self-play games of $\pi_1$, but also the data generated by pairing $\pi_1$ with any other policies in cross-play mode. It allows OBL to be trained on broader distributions to reduce side effects of the test time covariate shift.

## 2 Related Work

### 2.1 Prior Work on Zero-Shot Coordination

Zero-shot coordination (ZSC) was first introduced by Hu et al. [6] and further analyzed by Treutlein et al. [21] and led to a series of works meant to address the challenges of the setting. In their original paper, Hu et al. [6] introduced *other-play*, a method for training policies that are invariant with respect to the symmetries of the game, thus avoiding conventions that rely on arbitrary symmetry breaking. Other works have trained hierarchies of policies for ZSC [5], leveraged diverse populations to train a robust common best response [12, 19], and investigated the inductive bias of different architectures for leveraging action features [13]. One of the most significant breakthroughs in ZSC is off-belief learning (OBL) [7], which we detail in section 3.3 and upon which our work builds directly.

### 2.2 Belief learning

One crucial requirement for scaling OBL to large settings is learning beliefs about the private information of other players. Belief states can be explicitly tracked. Modern approaches to this problem involve deep sequence models, such as recurrent networks [20] or transformers [24]. However, the standard training setup for these models leverages Monte Carlo sampling. In contrast, OT-BL trains beliefs models via approximate dynamic programming. In this respect, OT-BL is most similar to the work of Sokota et al. [16], who propose a framework for belief fine-tuning. In this framework, a belief model for the current time-step is used to bootstrap a belief model for the next time-step using approximate dynamic programming. Sokota et al. [16] show that belief fine-tuning can improve the quality of belief models and adapt them to new policies at inference time. Belief fine-tuning and OT-BL are similar in that they use bootstrapping to learn beliefs; however, whereas Sokota et al. [16] examined belief fine-tuning in an inference-time context, we present OT-BL as a mechanism for performing off-team train-time learning.

# 3 Background

## 3.1 Turn-Based Dec-POMDPs with Public Actions

This work considers turn-based decentralized partially observable Markov decision processes (Dec-POMDP) [3] with public actions. Dec-POMDPs have state space $s \in S$, observation functions $o^i = \Omega^i(s)$ for each agent $i$ through which they obtain a partial observation of the underlying state, action space $A$, reward function $R: S \times A \to \mathbb{R}$ and transition function $\mathcal{T}: S \times A \to \Delta S$ that defines the distribution of next state given current state and joint action. At each time step, exactly one of $N$ players takes an action; this action is observable to all players. We further denote a trajectory as $\tau_t = (s_1, a_1, r_1, \cdots, s_t)$ and each players action observation history (AOH) $\tau_t^i = (o_1^i, a_1, r_1, \cdots, o_t^i)$. When $t$ equals to terminal $T$, the subscript is dropped and the two notations are simplified as $\tau$ and $\tau^i$ respectively. Each agent acts according to their policy $\pi^i(a|\tau_t^i)$ that returns a distribution over actions given action observation history and the joint policy is simply defined as $\pi = (\pi^1, \cdots, \pi^N)$. An important quantity in Dec-POMDPs is the belief function $\mathcal{B}_\pi(\tau_t|\tau_t^i) = P(\tau_t|\tau_t^i, \pi)$ that returns an agent's belief over the distribution of the state given their AOH and the group's joint policy.

## 3.2 Problem Settings

**Ad-Hoc Teamplay** The ad-hoc team play setting's goal is to cooperate effectively with unknown teammates, where each teammate can contribute meaningfully to the task [17]. This setting has been used in numerous environments including Overcooked [4], RoboCup 2D [2], and Hanabi [5, 7]. Some works leverage small amounts of data to guide agents towards equilibria or "social conventions" [11, 22]. For all experiments involving ad-hoc team play, we choose policies that are amenable to meaningful coordination.

**Zero-Shot Coordination and Intra-Algorithm Cross Play** Training in self play has been shown to give rise to policies that employ "arbitrary" conventions. Inspired by this, Hu et al. [6] proposed a zero-shot coordination (ZSC) in which the goal is to maximize intra-algorithm cross play (intra-AXP). Intra-AXP measures the cross play score between independently trained agents under the same algorithm.

**Human-AI Coordination** Multiple studies have evaluated AI agents in coordinating with humans, including in Hanabi [6, 15], Overcooked [4, 18]. One proxy setting for human-AI coordination is to evaluate with an agent trained on human data [4, 5, 7]. We use this proxy setting in this work.

## 3.3 Off-Belief Learning

**Intuition.** At every step, OBL *reinterprets* the AOH as if it is generated by playing with a fixed, given policy $\pi_0$. When $\pi_0$ is a fully random policy, past actions themselves are treated as conveying no information, since the probability of an action is independent of the trajectory.

**Implementation.** In practice, OBL trains a sequence of policies starting with $\pi_0$ a uniform random policy. Keeping track of the exact belief is computationally expensive. Thus, OBL is implemented by first training a neural network belief model on $\pi_0$ using supervised learning. This model is used to sample *fictitious trajectories* $\tau_t'$ by evaluating it on $\pi_1$'s data, i.e. $\tau_t' \sim \hat{\mathcal{B}}_{\pi_0}(\tau_t|\tau_t^i)$, where $\tau_t \sim P_{\pi_1}(\tau_t^i)$. At every timestep $t$, the policy $\pi_1$ is queried to produce an action on the *fictitious trajectory*. This action is used to perform a *fictitious transition*, which is used to calculate the target value for training OBL. See figure 2 for a visual representation of the fictitious transition.

# 4 Problem Setting

We define the off-team belief learning (OT-BL) problem setting as

$$\min_{\hat{\mathcal{B}} \in \Theta} \sum_t \sum_i \mathbb{E}_{\tau_t^i \sim P(\pi_b)} \mathcal{L}(\hat{\mathcal{B}}(\cdot \mid \tau_t^i), \mathcal{B}_{\pi_T}(\cdot \mid \tau_t^i)), \tag{1}$$

where $\Theta$ is the hypothesis class of belief models. In words, the goal of the off-team belief learning problem setting is to learn an accurate belief model $\hat{\mathcal{B}}$ for the policy $\pi_T$, where accuracy is weighted by the distribution over AOHs induced by a different policy $\pi_b$.

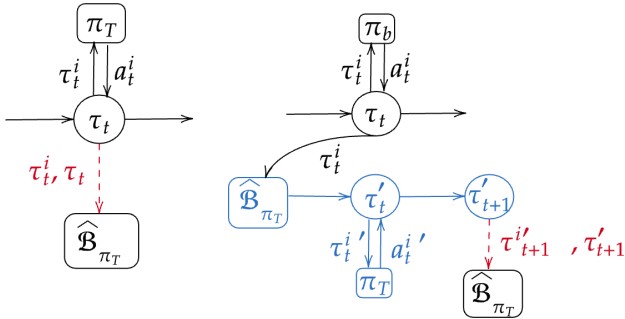

Figure 1: Visualization of belief training schemas by *vanilla OBL* (**left**) and *OT-BL* (**right**). Red arrows indicate values used for belief training. We only illustrate one time step, but we train on all time steps in a given trajectory, $\tau$. **Left**: *Vanilla OBL* trains $\hat{\mathcal{B}}_{\pi_T}$ to approximate the true conditional distribution $P(\tau_t|\tau_t^i)$ using trajectories $\tau$ sampled from $P_{\pi_T}$, independently for each time step $t$. **Right**: OT-BL uses $\pi_b$ to generate trajectories $\tau$ and trains $\hat{\mathcal{B}}_{\pi_T}$ on the distribution $P_{\pi_b}(\tau_t^i)$. In order to train $\hat{\mathcal{B}}_{\pi_T}$, we train only on fictitious transitions (indicated in blue) sampled from $\hat{\mathcal{B}}_{\pi_T}$ and use $\pi_T$ to generate the next action.

Similarly, we define the general OT-OBL problem setting as follows

$$\min_{\hat{q}\in\Theta}\sum_t\sum_i \mathbb{E}_{\tau_t^i,a_t\sim P(\pi_T^i,\pi_b^{-i})}\mathcal{L}\left(\hat{q}(\tau_t^i,a_t),\mathbb{E}_{\tau_t\sim P_{\pi_T}(\tau_t|\tau_t^i)}\left[R(\tau_t,a)+\mathbb{E}_{\tau_{t+1}\sim\mathcal{T}(\tau_t,a_t)}\hat{v}(\tau_{t+1})\right]\right),\tag{2}$$

where $\Theta$ is the hypothesis class of action-value functions and

$$\hat{v}(\tau)=\max_a\hat{q}(\tau^i,a)\tag{3}$$

where $i$ is the acting player. In words, the goal of the off-team off-belief problem setting learning is to learn an action value function $\hat{q}$ that has low OBL TD-error assuming that teammates played according to joint policy $\pi_T$ in the past, where error is weighted by the distribution over AOHs for agent $i$ is induced by the joint policy $(\pi_T^i,\pi_b^{-i})$.

## 5 Off-Team Learning

Off-team learning is a difficult problem because it requires weighting the loss values by the reach probabilities induced by $\pi_b$. However, we can't simply naively train our learned belief model, $\hat{\mathcal{B}}_{\pi_T}$, on data generated by $\pi_b$, since this would result in a belief model for $\pi_b$ rather than $\pi_T$. Similarly, we cannot simply naively train our policy on data generated by $\pi_b$ because such a policy would leverage the correlations between $\pi_b$'s actions and its private information. We give rough heuristics for approximating solutions for these problems under the names *off-team belief learning (OT-BL)* and *off-team off-belief learning (OT-OBL)*, which use bootstrapping to perform forward induction on the beliefs and backward induction the values (respectively), and trajectory resampling.

Say we have learned a belief model $\hat{\mathcal{B}}_{\pi_T}$ up to time $t-1$, and a value function $\hat{q}$ for time $t$ and thereafter. We describe off-team learning's procedure for generating data for time $t$ in three steps.

1. Off-team learning samples trajectories according to the behavior policy $\pi_b$, thereby guaranteeing that the samples $\tau_{t-1}^i$ are weighted by the behavior distribution.
2. Off-team learning corrects for the fact that $\tau_{t-1}$ was played to $t-1$ by $\pi_b$, rather than $\pi_T$. It does so by resampling $\tau_{t-1}'$ using the learned belief model, $\hat{\mathcal{B}}_{\pi_T}(\tau_{t-1}\mid\tau_{t-1}^i)$, taking advantage of the belief model being accurate up to time $t-1$.
3. We propagate this trajectory forward in time by sampling an action $a\sim\pi_T(\tau_{t-1}')$ from the target policy, $\pi_T$, and sampling a trajectory for the next time step $\tau_t'\sim\mathcal{T}(\tau_{t-1}',a)$ using this sampled action. This $\tau_t'$ is distributed approximately according to $\mathcal{B}_{\pi_T}(\tau_t^{i'})$. Furthermore, with the exception of the last action, which is generated with probability dictated by $\pi_b$. Finally, we train $\hat{\mathcal{B}}$ to maximize the likelihood of $\tau_t'$ given $\tau_t^{i'}$ and perform a TD-update between $\hat{q}(\tau_{t-1}^{i'},a)$ and $R(\tau_{t-1}',a)+\max_{a'}\hat{q}(\tau_t^{i'},a')$.

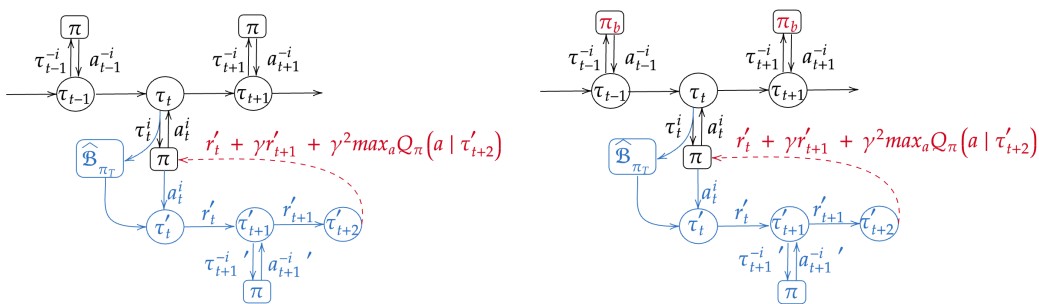

Figure 2: Visualization of policy training schemas for *vanilla OBL* (**left**), and *OT-OBL* on $P_{\pi, \pi_b}(\tau_t^i)$ (**right**) in a 2-player environment. Here superscript $i$ indicates the active player while $-i$ refers to their partner. The red arrow is the computed target value. The blue colors represent a *fictitious transition*. **Left**: *Vanilla OBL* generates trajectories by unrolling the training policy $\pi$ for both players. It then samples fictitious trajectories from $\tau_t' \sim \hat{B}_{\pi_T}(\cdot | \tau_t^i)$ to compute the training target. **Right**: OT-OBL instead generates data using $\pi$ for only one of the players and allows its partner to use a different policy $\pi_b$. It performs the same OBL training but only at time steps when $\pi$ is the active player.

These procedures are rough heuristic approximations for the problems described in the previous section but do not solve them exactly. We give pseudocode in Algorithms 1 and 2, and visualizations are provided in Figure 1 and Figure 2.

---

**Algorithm 1** OT-BL: Train a belief model $\hat{B}_{\pi_T}$ for target policy $\pi_T$ that is robust when evaluated on behavioral policy $\pi_b$ (i.e. to mitigate out-of-distribution errors when used to reinterpret trajectories generated by $\pi_b$). The main difference between OT-BL and normal belief training is marked in red.

1: **procedure** OT-BL($\pi_T$, $\pi_b$, $d$)
   ▷ $\pi_T$: the target policy for which we want to train a belief model.
   ▷ $\pi_b$: the behavioral policy on which the belief model may be evaluated at test time.
   ▷ $d$: size of the dataset
2:     Initialize belief model $\hat{B}_{\pi_T}$
3:     Initialize dataset $\mathcal{D} = \emptyset$
4:     Initialize environment $\tau = \tau_0 \sim P(\tau_0)$
5:     **while** len($\mathcal{D}$) $< d$ **do**
6:         Sample fictitious trajectory $\tau_t' \sim \hat{B}_{\pi_T}(\cdot | \tau_t^i)$
7:         Unroll the fictitious trajectory $\tau_{t+1}' = \mathcal{T}(\tau_t', a_t')$ with target policy $a_t' \sim \pi_T$
8:         Add training data from the unrolled fictitious trajectory $(\tau_{t+1}'^i, \tau_{t+1}')$ to dataset $\mathcal{D}$
9:         Unroll the real trajectory $\tau_{t+1} = \mathcal{T}(\tau_t, a_t)$ with behavioral policy $a_t \sim \pi_b$
10:        Reset $\tau = \tau_0 \sim P(\tau_0)$ **if** $\tau_{t+1}$ is terminal
11:    **end while**
12:    **repeat**
13:        Train $\hat{B}_{\pi_T}$ with loss: $-\log p(\tau_t | \tau_t^i)$ witht mini-batch from dataset $\{\tau_t^i, \tau_t\} \sim \mathcal{D}$
14:    **until** $\hat{B}_{\pi_T}$ converges
15: **return** $\hat{B}_{\pi_T}$
16: **end procedure**

---

# 6 Experimental Setup

## 6.1 Hanabi

Hanabi [1] is a benchmark for Dec-POMDP research. It is a cooperative game for 2-5 players played with 8 hint tokens, 3 life tokens, and a deck of 50 cards, each having a rank between 1 and 5 and one of five colors. In the 2-player variant, on which this paper focuses, each player holds 5 cards, backwards, such that they can see their partners' hand but not their own. On their turn, a player can either *play*, *discard* or *hint*. The goal is to play cards in rank order for each color, with each correctly played card providing one point for a maximum score of 25. The team loses a life token each time

---

**Algorithm 2** OT-OBL: Train a Q-function under the OBL setting, but also make it less prone to out-of-distribution errors when evaluated with different partners such as $\pi_b$. For simplicity, we show the procedure for data collection and policy update with a single transition, omitting the complicated distributed RL setup. We assume there are two players, $i$ (current) and $-i$ (next). The main difference between OT-OBL and normal OBL is marked in red.

1: **procedure** OT-OBL($\hat{q}$, $\pi$, $\hat{\mathcal{B}}_{\pi_T}$, $\pi_b$)
    ▷ $\hat{q}$, $\pi$: the Q-function we are training and its associated (greedy) policy
    ▷ $\hat{\mathcal{B}}_{\pi_T}$: The belief model of the input policy $\pi_T$ ($\pi_0$ in original OBL)
    ▷ $\pi_b$: the policy that may potentially be a partner at test time
    ▷ Note: assume that player $i$ uses policy $\pi$ and $-i$ uses policy $\pi_b$
2:      Generate a $t$-step partial trajectory $\tau_t \sim P_{(\pi,\pi_b)}(\tau_t)$ using joint policy $(\pi, \pi_b)$, $t \geq 0$.
3:      Sample fictitious trajectory $\tau'_t \sim \hat{\mathcal{B}}_{\pi_T}(\cdot|\tau_t^i)$
4:      Unroll the fictitious trajectory $\tau'_{t+1} = \mathcal{T}(\tau'_t, a'_t)$ with $a'_t \sim \pi(a|\tau_t^i)$, get reward $r'_t$
5:      Unroll the fictitious trajectory $\tau'_{t+2} = \mathcal{T}(\tau'_{t+1}, a'_{t+1})$ with $a'_{t+1} \sim \pi(a|\tau'^{-i}_{t+1})$, get reward $r'_{t+1}$
6:      Update Q-function $\hat{q}(a'_t|\tau_t^i) \leftarrow r'_t + \gamma r'_{t+1} + \gamma^2 \max_a \hat{q}(a|\tau'^i_{t+2})$
7: **end procedure**

---

a player attempts to play an unplayable card. Hinting consumes a hint token and can be used to point all cards of a chosen color or rank in a partner's hand[3]. The team regains a hint token when a player discards a card or completes a stack by successfully playing the 5 of the corresponding color. A player draws a new card after playing or discarding. The game terminates when all 5 stacks are completed, the deck is empty, or all three lives are lost. In the latter case the team loses all the points they have earned.

## 6.2 Training Details

We implement our algorithms based on the open-sourced code for OBL [4] and extend it with ideas from synchronous training [10, 5], training all models simultaneously, thus enabling effective implementations of OT-BL and OT-OBL. When synchronously training OBL models, every $n = 50$ steps we save a copy of the model then query for and update all dependencies.

We follow practices in the original OBL paper [7] to train each policy and belief model. The backbone is R2D2 [8], a distributed recurrent Q-learning method with parallel environment workers and centralized replay buffer and trainer. We briefly describe the loss functions for the Q-network and belief models here. For more details, please refer to Appendix A.1 or the OBL paper [7].

The Q-network is trained to minimize the TD-error between its Q-value and targets from fictitious transition: $\mathcal{L}(\boldsymbol{\theta}|\tau) = \frac{1}{2}\sum_{t=1}^{T}\left[G'_t - Q_{\boldsymbol{\theta}}(a_t|\tau_t)\right]^2$ where the target $G'_t$ is calculated on the 'imagined' fictitious transitions $G'_t = r'_t + \gamma r'_{t+1} + \gamma^2 \max_a Q_{\hat{\boldsymbol{\theta}}}(a|\tau'_{t+2})$ and $\hat{\boldsymbol{\theta}}$ is a slightly outdated target network. We pre-compute the fictitious target $G'_t$ and store them into the replay buffer to avoid branching LSTM rollouts into fictitious transitions at training time.

The belief models are trained to predict the player's hand given $\tau^i$. At a given timestep $t$, the belief model will take in $\tau_t^i$ as input and predict the hand of player $i$ auto-regressively from oldest card to newest. We train all OBL models as recurrent models via supervised learning to optimize $\mathcal{L}(\boldsymbol{h}|\tau_t^i) = -\sum_{k=1}^{n} \log p(h_k|\tau_t^i, h_{1:k-1})$ where $h_k$ is the $k$-th card in hand and $n$ is the hand size.

## 6.3 Evaluation

We repeat each training method five times with different seeds. For SP and XP, every team is evaluated on 5000 games and then averaged over five seeds. The XP score is the average score of all teams formed by all independently trained agents. To better reflect improvement in intra-AXP, we introduce the *self-play cross-play gap (SXG)* metric, which is computed as SXG $= 100 \cdot \frac{(SP-XP)}{SP}$. This

---

[3]e.g. "Your first and third card are rank 3."
[4]https://github.com/facebookresearch/off-belief-learning

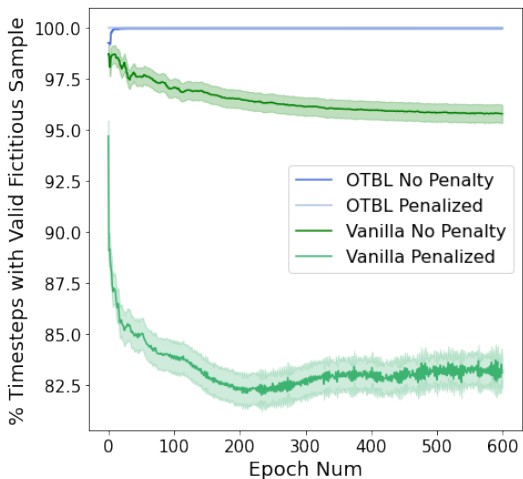

Figure 3: Comparison between belief models trained with vanilla OBL (greens) and with OT-BL (blues) in 2-life Hanabi with and without penalty for invalid samples. Experiments with and without penalty sample $p = 1, 10$ hands respectively. We plot the % of timesteps with a valid fictitious sample (standard error is shaded) when training OBL level 1. Normally trained belief models struggle to gather a valid sample; it steadily declines in unpenalized cases and drops to only $\sim 83\%$ of timesteps in the penalty cases. OT-BL is able to consistently gather a valid sample during training ($> 99.5\%$ of the time), regardless of penalty. OT-BL mitigates the covariate shift experienced during OBL training.

normalizes increases in XP score relative to the theoretical maximum self-play score. Additionally, SXG acts as a proxy for the covariate shift as it compares the evaluated training distribution (SP) with the test distribution (XP). The higher the SXG the larger the covariate shift.

## 6.4 Experimental Design for OT-BL

We design experiments for OT-BL to emphasize its advantage over the normal belief learning procedure when evaluated off-team. In normal OBL, the belief model is trained on trajectories from $\pi_\ell$ but utilized on trajectories from $\pi_{\ell+1}$ to produce fictitious transitions, making it vulnerable to covariate shift. With OT-BL, half of environments used to train $\hat{\mathcal{B}}_{\pi_\ell}$ follow typical belief model training. The other half uses OT-BL to train $\hat{\mathcal{B}}_{\pi_\ell}$ off-team on $P_{\pi_{\ell+1}}(\tau_t^i)$.

Ideally, we want to evaluate our algorithm on environments with a *larger covariate shift*. Therefore, we make a few modifications to the Hanabi environment to exacerbate the covariate shift. First, we reduce the total number of life tokens from 3 to 2, making the game harder so that the average score and average trajectory length of different OBL levels differ more significantly. It is particularly challenging for the belief model only trained on shorter trajectories to generalize to longer unseen trajectories. Second, we consider *sampling valid hands*. Prior works including OBL utilize a checker provided in the Hanabi challenge [1] to check if a hand satisfies the known public knowledge. If a hand is invalid, the environment can fail to produce key input features. We further modify the 2-life Hanabi environment to remove the checker and allow invalid samples to corrupt the environment as is, which only affects the fictitious transition. Specifically, if a sampled card contradicts the public knowledge of that card, the observation encoder will just encode the card knowledge for that card to be all zero instead of rejecting the sampled hand. We call this *invalid card penalty*.

Finally, we measure the belief improvements through the percentage of valid samples for fictitious transitions. For this we focus on the level 1 policy training process in the OBL hierarchy, where there is the largest covariate shift. To train the level 1 policy, we need to unroll a belief model $\hat{\mathcal{B}}_{\pi_0}$ on trajectories generated by the learning policy $\pi_1$. At convergence, $\pi_1$ can achieve 20 points. However, the belief model is trained on $\pi_0$, a uniform random policy with average score close to zero. We compute the percentage of timesteps where $\hat{\mathcal{B}}_{\pi_0}$ is able to produce a valid sample when training $\pi_1$ as a proxy for the amount the belief model has deteriorated due to covariate shift.

| Level | without penalty | | with penalty | |
| --- | --- | --- | --- | --- |
| | OBL (XP & SXG) | OT-BL (XP & SXG) | OBL (XP & SXG) | OT-BL (XP & SXG) |
| 1 | $20.17 \pm 0.15$ (0.84) | $21.05 \pm 0.02$ (0.05) | $16.71 \pm 0.06$ (1.56) | $21.15 \pm 0.02$ (0.09) |
| 2 | $22.92 \pm 0.21$ (2.18) | $23.41 \pm 0.02$ (0.56) | $21.24 \pm 0.11$ (4.28) | $23.48 \pm 0.02$ (0.38) |
| 3 | $23.05 \pm 0.30$ (3.27) | $23.76 \pm 0.02$ (0.46) | $22.53 \pm 0.23$ (4.43) | $23.83 \pm 0.01$ (0.33) |

Table 1: XP and SXG (in parentheses) for OBL and OT-BL in 2-life Hanabi with and without *invalid card penalty*. OT-BL always achieves higher XP scores. OT-BL also has a very small SXG $< 1\%$ which is $4 - 18\times$ lower than corresponding vanilla OBL. The $\pm$ represents the team's standard error.

| Level | OBL (XP & SXG) | OT-OBL(XP & SXG) | OT-OBL+ Robust Belief (XP & SXG) |
| --- | --- | --- | --- |
| 1 | $21.50 \pm 0.05$ (0.42) | $21.31 \pm 0.01$ (0.19) | $21.61 \pm 0.03$ (0.32) |
| 2 | $23.75 \pm 0.02$ (0.46) | $23.72 \pm 0.01$ (0.17) | $23.78 \pm 0.01$ (0.21) |
| 3 | $24.10 \pm 0.02$ (0.41) | $24.10 \pm 0.01$ (0.17) | $24.22 \pm 0.01$ (0.17) |
| 4 | $24.03 \pm 0.10$ (0.91) | $24.22 \pm 0.01$ (0.21) | $24.29 \pm 0.01$ (0.16) |
| 5 | $23.99 \pm 0.11$ (1.28) | $24.24 \pm 0.01$ (0.12) | $24.30 \pm 0.01$ (0.16) |

Table 2: XP and SXG (in parentheses) for vanilla OBL, OT-OBL, and OT-OBL with belief models trained on broader distributions in 3-life Hanabi. Vanilla OBL overfits leading to a drop in XP scores. OT-OBL prevents overfitting and continues to improve its XP score. OT-OBL almost always has a higher XP scores than vanilla OBL. OT-OBL has a very small SXG $< 1\%$ and this gap is up to 10 times lower than that of vanilla OBL. By training our belief models on broader distributions, we further increase our XP score while maintaing a low SXG. The $\pm$ represents the team's standard error.

## 6.5 Experimental Design for OT-OBL

**OT-OBL for Intra-AXP** We design experiments to evaluate OT-OBL's performance in intra-AXP. For normal OBL, the policy, $\pi_\ell$, is trained on trajectories from $P_{\pi_\ell}(\tau_t^i)$ but evaluated on $P_{\pi_\ell, \pi_b}(\tau_t^i)$. This means $\pi_\ell$ experiences test time covariate shift. To evaluate OT-OBL in intra-AXP, when training a given policy $\pi_\ell$, we train on real trajectories from from $P_{\pi_{\ell-1}, \pi_\ell}(\tau_t^i), P_{\pi_\ell}(\tau_t^i), P_{\pi_\ell, \pi_{\ell+1}}(\tau_t^i)$ and use OT-OBL to train off-team. By training on broader distributions, we increase the support of our policies. Since $\pi_{\ell-1}$ and $\pi_{\ell+1}$ are OBL policies, they are also reasonable and do well in Hanabi. We also test a variant where we use OT-OBL and robustify the belief by also training the belief model on $P_{\pi_{\ell-1}, \pi_\ell}(\tau_t^i), P_{\pi_\ell}(\tau_t^i), P_{\pi_\ell, \pi_{\ell+1}}(\tau_t^i)$. We refer to this variant as OT-OBL + robust belief.

**OT-OBL for ad-hoc teamplay and human-AI coordination** This set of experiments aims to demonstrate the ability for OT-OBLto improve test time performance with only a small amount of data from test time agents. These experiments address the ad-hoc teamplay and human-AI coordination settings where there are *large covariate shifts*. We utilize *rank bot* for the ad-hoc teamplay setting and *clone bot*, a bot trained via supervised learning on human data, as a proxy for human-AI coordination. See Appendix A.3 and A.4 for more details on training *rank* and *clone bot* respectively. Each setting we assume access to only a small number, 3200, of full trajectories generated by the target agents. We retrain a new policy, $\pi$, with a belief model trained on level 3 of sequentially trained OBL with the typical OBL training procedure. In this case half of the games in our distributed RL environment are the real target trajectories and we train $\pi$ in an off-policy manner with OT-OBLon these trajectories. By training on these target trajectories, we can partially mitigate the covariate shift experienced when $\pi$ is evaluated. We compare our results with a vanilla sequentially trained OBL level 4.

# 7 Results and Discussion

## 7.1 OT-BL

The the XP and SXG results of OT-BL on 2-life Hanabi is shown in Table 1. Without the *invalid card penalty*, both OBL with vanilla belief learning and OT-BL are able to achieve reasonably high XP scores in 2-life Hanabi. However, OT-BL consistently achieves higher XP scores in 2-life Hanabi across all levels. At level 3, OT-BL is able to achieve a final XP score of $23.76 \pm 0.02$. Lastly, the SXG for OT-BL is always lower than that of vanilla OBL (8-16x lower).

| Training Type | XP & SXG | w/ rank bot | w/ clone bot | w/ Vanilla OBL |
|---|---|---|---|---|
| Vanilla OBL | $23.66 \pm 0.06$ (1.42) | $14.47 \pm 0.67$ | $16.19 \pm 0.21$ | - |
| OT-OBL w/ Rank Bot | $23.59 \pm 0.08$ (1.38) | $15.66 \pm 0.56$ | $16.36 \pm 0.23$ | $23.67 \pm 0.04$ |
| OT-OBL w/ Clone Bot | $23.54 \pm 0.06$ (1.13) | $14.33 \pm 0.63$ | $16.64 \pm 0.20$ | $23.64 \pm 0.04$ |

Table 3: Evaluating OT-OBL in ad-hoc teamplay (w/ rank bot), proxy human-AI coordination, and w/ vanilla OBL. By training on a small number of target trajectories, we can improve ad-hoc teamplay scores with rank bot and proxy human-AI coordination scores. OT-OBL also doesn't significantly hurt XP scores, the SXG, or the unseen target policies in ad-hoc teamplay or human-AI coordination. Additionally, by training with OT-OBL, we still achieve very similar scores with vanilla OBL, indicating we are only broadening the policies support. The $\pm$ here represents the team's standard error.

In Table 1, we also show the performance of OT-BL on 2-life Hanabi *with invalid card penalty* on the right. Similarly, OT-BL consistently achieves higher XP scores across all levels. OT-BL is able to achieve a small SXG (always $< 1\%$) and this gap is up to $18\times$ lower than vanilla OBL. OT-BL achieves similar results between the penalized and unpenalized environments since the belief model is more on distribution, thus it almost always samples valid samples ($> 99.5\%$ of the time). However, OBL with vanilla belief learning suffers significantly from the penalty, especially for level 1 where the covariate shift from a uniform random policy to a learned OBL policy is the largest. This indicates that the test time covariate shift is lower for OT-BL when compared to vanilla OBL. The reduced covariate shift improves training and leads to better test time results.

We plot the percentage of timesteps $\hat{\mathcal{B}}_{\pi_0}$ is able to produce a valid sample when training $\pi_1$ in Figure 3, to illustrate the deteriorating belief model in normal OBL and show the benefit of OT-BL. *Without penalty*, vanilla OBL steadily decreases in the percentage of timesteps producing valid samples. *With penalty*, vanilla OBL is unable to produce a high percentage of valid samples and can never recover. This high percentage of invalid samples contributes to the steep score drop for OBL level 1 in the invalid state experiments. On the other hand, OT-BL mitigates the belief model's training covariate shift and consistently samples valid states (over $99.5\%$ of the time). This contributes to the high scores for OT-BL and explains why OT-BL maintains high scores in penalized Hanabi.

## 7.2 OT-OBL

**Intra-AXP Results** We present the XP and SXG results of OT-OBL in table 2. All variants, normal OBL, OT-OBL , and OT-OBL + robust belief all achieve high XP scores. However, at level 4 vanilla OBL starts to overfit, whereas OT-OBL and OT-OBL + robust belief continue to improve. This results in OT-OBL achieving a XP score of $24.24 \pm 0.01$ and OT-OBL + robust belief achieves a XP score of $24.30 \pm 0.01$. As Vanilla OBL overfits, the SXG increases, indicating an increase in the test time covariate shift. On the other hand, both OT-OBL and OT-OBL + robust belief always have a SXG$< 1\%$ and at level 5 the SXG is almost 0. By level 5, the SXG of OT-OBL and OT-OBL + robust belief are at least 8 times lower than that of vanilla OBL. Overall, by training on broader distributions, we reduce the test time covariate shift, achieve goof intra-AXP performance, and improve OBL in off-team evaluation.

**Ad-Hoc Teamplay and Proxy Human-AI Coordination Results** We present the results appling OT-OBL Ad-hoc teamplay and Human-AI coordination in table 3. With OT-OBL and only 3200 target policy trajectories, we can increase *rank bot* to $14.47 \pm 0.67$ to $15.66 \pm 0.56$ and increase *clone bot* scores from $16.19 \pm 0.21$ to $16.64 \pm 0.20$ without heavily impacting XP scores or the SXG. Additionally, vanilla OBL in XP achieves near identical scores to OT-OBL trained on additional trajectories is paired with vanilla OBL. Therefore, the additional trajectories merely broaden the training policies support and don't negatively impact policy training. Overall, by training on target distributions off-policy and off-team with OT-OBL we are able to reduce the test time covariate shift and increase ad-hoc teamplay and proxy human-AI coordination scores.

**Failure Cases Bombing Out** Bombing out is an important failure case in Hanabi, where if three unplayable cards are played then team loses all of their points and the game ends prematurely. In Hanabi, an agent always has alternatives to playing a card (e.g. discarding or hinting), thus a bombout occurs when an agent misunderstood the hints given by a partner or if its Q-value predictions are inaccurate. Both of which can be caused by covariate shifts. We present bombing out results in the

| Cross-play Pair | Total number of bombouts | Number of bombouts by OBL/OT-OBL | % Bombouts by OBL/OT-OBL |
|---|---|---|---|
| Vanilla OBL × Rank Bot | 6891 | 1619 | 23.49% |
| OT-OBL w/ Rank × Rank Bot | 6048 | 1208 | 19.97% |
| Vanilla OBL × Clone Bot | 6144 | 663 | 10.79% |
| OT-OBL w/ Clone × Clone Bot | 5239 | 473 | 8.28% |

Table 4: Evaluating total number of bombouts and percentage of bombouts by vanilla OBL or OT-OBL in ad-hoc teamplay with two different partners. Each pair is first evaluated with 5000 games and then averaged over 5 independently trained OBL or OT-OBL models. By training with OT-OBL, we are able to significantly reduce instances where our trained agent causes bombouts.

ad-hoc teamplay and proxy-human AI coordination setting in table 4. Notice, that OT-OBL reduces the instances where our agent causes bombouts by 25.38% when partnered with rank bot and 28.66% when partnered with clone bot.

## 8 Limitations

The largest set of limitations of the approaches proposed in this work is that they are only heuristic approximations for solving the problems that this work introduced and do not converge to the exact solutions. Additional limitations include the fact that OT-BL assumes $\pi$ is also on distribution when we produce fictitious moves with OT-BL. As a result, if $\pi$ is too off distribution OT-BL might not mitigate covariate shift. Moreover, although our work provides a positive step to mitigate the large covariate shift that can occur in ad-hoc teamplay and human-AI coordination with little target policy data, there is still exists covariate shift.

## 9 Conclusion

We present *off-team belief learning*, a method that allows us to approximate training beliefs *off-team*. Belief models are very ubiquitous in POMDPs and Dec-POMDPs. However, they can experience covariate shift when deployed on unseen distributions. We demonstrate that standard belief model training can suffer from covariate shift, but OT-BL reduces it. Overall, by training OBL with OT-BL, we are able to achieve strong results in intra-AXP in 2-life Hanabi and minimize the gap between SP and XP.

We also present *off-team off-belief learning*, a method that allows us to approximate train action values functions *off-team*. The goal of off-team training is to train with various teams with the goal of improving performance with unseen teams. With OT-OBL, we can train policies on broader distributions, increase their support, and improve test time performance. By training OBL with OT-OBL we were able to achieve near-optimal intra-AXP performance and improve ad-hoc teamplay and proxy human-AI coordination.

## 10 Broader Impact

We do not believe this work raises broader impact concerns.

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
