# A  Experimental Details

## A.1  Policy Training

As mentioned, we implement R2D2 a large scale deep-Q learning method [8]. Our implementation of R2D2 has 6400 inference environments running in parallel. We achieve this by running 80 threads and 80 environments per thread. Every agent is a neural network on GPU and selects an action via GPU calls. We dynamically batch model calls onto the GPU in order to increase inference speed. By running multiple environments in a thread, we can forward non-blocking environments while others are waiting for model calls. Additionally, running environments in parallel saturates GPU usage. When a trajectory terminates, the environment loop grabs all necessary actions, observations, and targets, pads everything to a length of 80 and adds it to the centralized replay buffer as done in [14].

At every step in training, we sample a batch of trajectories from the prioritized replay buffer and train on the TD-error of the trajectory. We then update the priority of the trajectory in the replay buffer as $\xi = 0.9 \cdot \max_i \xi_i + 0.1 \cdot \xi$ [8], where $\xi_i$ is the TD error per step. The training loop is consistently running and sampling from the replay buffer. After 10 training updates, the training loop updates the inference policies. For OBL, there are dependencies between policy and belief training. For example training $\hat{\mathcal{B}}_{\pi_\ell}$ relies on $\pi_\ell$ and similarly training $\pi_\ell$ relies on $\hat{\mathcal{B}}_{\pi_{\ell-1}}$. To account for this we query for an updated version of the dependency every 50 training steps as done in [10, 5]. For trajectory generation, we use epsilon greedy exploration at training time. At the beginning of every simulated game generate $\epsilon_i$ from the equation $\epsilon_i = \alpha^{1+\beta \frac{i}{N-1}}$, where $\alpha = 0.1, \beta = 7, N = 80$.

The entire inference and training infrastructure for a single policy or belief model uses a machine with 30 CPU cores and 2 GPUs, one GPU for training and one for simulation. Our infrastructure design allows us to only re-use trajectories 4-5 times before it is evicted from the replay buffer.

We use the same policy network architecture from [7]. We use their public-lstm architecture design. We use a 3-layer feedforward neural network to encode the entire private observation. The public observation is encoded with a one-layer feedforward neural network followed by a LSTM. We combine the private and public encoding via element-wise multiplication. As done in [8] we use a dueling architecture [25] to produce the final Q-values and use double DQN [23].

For all OBL experiments, we only compute the target based on $m = 1$ fictitious steps. Additionally, unless otherwise noted at every timestep $t$, we sample from the belief model $p = 10$ times, unless otherwise noted, and use the first sample that doesn't violate card constraints.

We present other relevant hyperparameters for policy training in table 5.

| Hyper-parameters | Value |
|:---:|:---:|
| *Optimization Parameters* | |
| optimizer | Adam[9] |
| lr | $6.25e - 05$ |
| eps | $1.5e - 5$ |
| gradient clip | 5 |
| batchsize | 128 |
| *Q-learning Parameters* | |
| discount factor | 0.999 |
| num gradient steps sync target net | 2500 |
| *Replay Buffer Parameters* | |
| burn-in-frames | 10000 |
| replay buffer size | $131072\ (2^{17})$ |
| priority exponent | 0.9 |
| priority weight | 0.6 |

Table 5: Hyper-parameters for Hanabi agent training

## A.2 Belief Training

Belief training follows a similar procedure to policy training; this has also been done by [7]. Inference workers populate a centralized replay buffer, which is sampled from and used to train our belief model. Typically, if OBL is sequentially trained e.g. each belief/policy is trained to convergence before being used, then the policy used for belief training is fixed. However, this changes since we are synchronously training our OBL hierarchies. Specifically, in standard OBL when training $\hat{\mathcal{B}}_{\pi_\ell}$ we train on $P_{\pi_\ell}(\tau_t^i)$, but $\pi_\ell$ is being concurrently trained. Thus, similar to policy training, every $50$ gradient steps, we query for a newly updated $\pi_\ell$.

There are differences between belief and policy training. First, we store the desired true hand of the player alongside the observation, which will be used to train the belief model. Second, we use a normal experience replay buffer, e.g. one without priority. Lastly, the training loop trains an autoregressive belief model that is trained to predict cards one-by-one from oldest to newest via supervised learning.

Typically, our belief models use a standard LSTM and the associated hidden state as the recurrent model. However, in order to adjust for algorithmic changes and allow for fast unrolling of our LSTM, in experiments involving OT-BL or comparisons performed with standard OBL, the hidden state at timestep $t$ is modified to be:

$$h_{\text{ffwd}}^t = \text{ffwd}(\tau_t^{j\prime})$$
$$h_{\text{lstm}}^t = \text{lstm}(\tau_{t-1}^j)$$
$$h = \text{ffwd}_{\text{joint}}(\text{concat}([h_{\text{ffwd}}, h_{\text{lstm}}]))$$

**Belief Model Accuracy** The belief model is used to resample the true state, which means samples drawn from it must 1) not violate the agent's private observations and 2) match the distribution over states given the trajectory so far. From a theoretical standpoint, for 1), it suffices to not train on those samples, either by resampling until we obtain a valid state or rejecting the transition from the replay buffer. Property 2) follows from forward induction, and we provide a rough sketch here:

At time $t = 0$, before agents take any action, all belief models are accurate as long as their sample distribution matches the environment distribution of initial states. Assuming the belief is accurate up to time $t - 1$ for policy $\pi$, we can freeze the policy and train the belief on step $t$ until convergence. We then update the policy slightly, freeze it again, and adjust the belief to be accurate w.r.t. the updated policy. The belief will now be accurate up to step $t$.

Empirically, we do not need to freeze the models since we found the belief model to train much faster (and if it didn't, we could simply slow down policy training artificially), such that policy training always occurs off a sufficiently up-to-date belief. Furthermore, in our setting, trajectories of untrained policies tend to be short since they terminate the game by bombing out. Thus, as reported in Figure 3, OT-BL can draw valid samples > 99% of the time in 2-life Hanabi with virtually no decline as the policy is updated. This, paired with the relative training speed, shows that we can readily maintain an accurate belief model when jointly training the belief and policy in the large scale complex environment Hanabi.

## A.3 Details on Rank Bot

We train a bot that responds effectively to rank hints, a very common convention in Hanabi. This policy uses the same network design as all of our other policies.

Rank bot is trained with other-play (OP) [6]. In the two player setting, OP has a player observe the world from a color permuted space, e.g. the observation and action spaces follow the same color permutation. The color permuation is sampled once at the beginning of an episode. In Hanabi, color is a well-known symmetry. By shuffling colors it prevents agents from forming arbitrary symmetry breaking conventions. Previously, bots trained with OP achieved the best ZSC score in Hanabi. Empirically, agents trained with OP use rank to hint a playable card to their partner. The agent receiving the hint tends to play its newest card that matches the rank hint (even if it doesn't know the card's color).

### A.4 Details on Clone Bot

The longstanding goal of cooperative MARL is to cooperate effectively with novel agents at test time. To approximate human interaciton in Hanabi, we used a dataset of 240,954 games obtained from Board Game Arena (https://en.boaradgamearena.com/). We converted these games to a dataset for supervised learning. At every timestep $t$, the model takes in $\tau_t^i$ and predicts the action. We optimize the network via cross-entropy loss. The network is built from one fully connected layer, which is fed into a ReLU activation, and then into a two-layer LSTM. This output is fed into a fully connected layer plus a softmax. We apply dropout before the last layer to reduce overfitting. The models use 512 hidden units per layer and a dropout rate of 0.5. We also use color shuffling from [6] as a data augmentation technique. Every training step, we randomly shuffle the color space of the observation and action before training. At test time we select the agent that achieves the highest self-play score during training and use it as "Clone Bot."