# OpenReview forum: "Off-Team Learning"
_NeurIPS.cc/2022/Conference — NeurIPS 2022 Accept_

### Official Review · Reviewer_nA7t · 2022-06-26

**Rating:** 6
**Confidence:** 3
**Soundness:** 3 good
**Presentation:** 2 fair
**Contribution:** 2 fair

**Summary:**

This paper investigates the problem of off-policy learning in cooperative games. Within this area, they address a problem they call "off-team (OT) learning", where the teammates that generated the experience differ from the target teammates. They accomplish this by disentangling the belief in the trajectory, specifically the previous joint actions, from action evaluation. As a result, a policy can be learned that may adapt to a target team through its belief maintenance. Their algorithm is evaluated in a variety of Hanabi settings and shows some empirical improvements.

**Questions:**

1. "Off-team learning" relies on the learned belief model bein accurate up to time t-1. This seems unrealistic in practice to rely on as an assumption when the algorithm is jointly learning it alongside the policy. Can the authors comment on the need for accuracy in the belief model both in terms of the theoretical guarantees of the algorithm and the empirical performance?

2. Coordination games may exhibit many diverse equilibria. How does this algorithm cope with target teammates that play equilibrium vastly different than that expressed by the behavioural teammates?

3. The authors claim L224: "The goal of ZSC is to evaluate the consistency of an algorithm across different seeds." This is not true. The goal of ZSC is to coordinate productively with unseen opponents. If the authors' statement was true any deterministic algorithm solves this problem.

4. In L236 the authors discuss the robustness of their algorithm given covariate shift with respect to the environment dynamics. However, it seems to me, that is mostly orthogonal to the work. In the abstract it seems the focus is on only differing teammates. Shouldn't the emphasis be on covariate shift to radically different teammates? And/or the combination of the two changes? Could the authors explain this focus and why they de-emphasized these other facets?

5. L283, could the authors explain "only a small number, o=3200, of trajectories"? How many trajectories are 3200 observations? Why is it measured in observations?


**Limitations:**

The broader impact section could be improved to reflect the impact belief-modelling style algorithms can have on biases.

**Strengths And Weaknesses:**

**Strengths**
 - Their algorithms are described in the text, graphics, and pseudocode allowing for diverse learning styles to approach the material.
 - Evaluations include some amount of ablations/sensitivity-analysis to better understand the merits of the proposed algorithm.

**Weaknesses**
 - The paper is dense with jargon and heavily marked symbols with detailed meanings making it more challenging to read than I believe is necessary.
 - Maintaining and correcting for beliefs as a way to make strategic adjustments has been in many flavors, making this work's originality small.

---

> ### Author Response · Authors · 2022-08-02
> **Reply to Reviewer nA7t**
>
> We thank the reviewer for their constructive comments and feedback.
>
> We first wanted to ask for clarification from the reviewer on their statement “Maintaining and correcting for beliefs as a way to make strategic adjustments has been in many flavors, making this work's originality small.” Could the reviewer point us to references that maintain and correct beliefs in multi-agent reinforcement learning?
>
> We also address comments below:
>
> **Accuracy of belief model**
>
> The belief model is used to resample the true state, which means samples drawn from it must 1) not violate the agent's private observations and 2) match the distribution over states given the trajectory so far.
> From a theoretical standpoint, for 1), it suffices to not train on those samples, either by resampling until we obtain a valid state or rejecting the transition from the replay buffer. Property 2) follows from forward induction, and we provide a rough sketch here:
>
> At time $t=0$, before agents take any action, all belief models are accurate as long as their sample distribution matches the environment distribution of initial states. Assuming the belief is accurate up to time $t-1$ for policy $\pi$, we can freeze the policy and train the belief on step $t$ until convergence. We then update the policy slightly, freeze it again, and adjust the belief to be accurate w.r.t. the updated policy. The belief will now be accurate up to step $t$.
>
> Empirically, we do not need to freeze the models since we found the belief model to train much faster (and if it didn't, we could simply slow down policy training artificially), such that policy training always occurs off a sufficiently up-to-date belief. Furthermore, in our setting, trajectories of untrained policies tend to be short since they terminate the game by bombing out. Thus, as reported in Figure 3, OT-BL can draw valid samples > 99% of the time in 2-life Hanabi with virtually no decline as the policy is updated. This, paired with the relative training speed, shows that we can readily maintain an accurate belief model when jointly training the belief and policy in the large scale complex environment Hanabi.
>
> **Diverse equilibria in coordination games**
>
> If the equilibrium shift is larger, then it leads to a larger covariate shift. OT-RL helps broaden the training distribution of the behavior policy, allowing for better performance with the target policy (with limited trajectories from the target policy).
>
> **Goal of zero-shot coordination (ZSC), a deterministic algorithm will solve ZSC**
>
> We use the zero-shot coordination definition as introduced in Hu et al. 2020:
>
> “Our setting is a partially observed cooperative Markov game (MG) which is commonly known among both agents. The agents are able to construct strategies separately in the training phase but cannot coordinate on the strategies that they construct. They must then play these strategies when paired together one time. We refer to this as zero-shot coordination.”
>
> Under this definition, further detailed in section 3 of Hu et al. 2020, agents will play a common game, but cannot pre-coordinate their strategies. They can however coordinate on a learning rule prior to receiving the Dec-POMDP specification. Thus, ZSC does not require coordination with any "unseen partners"; it only concerns itself with partners trained independently with the same algorithm. This is not solvable by deterministic algorithms because it is impossible to specify deterministic preferences before seeing the task (and underlying reward structure). Moreover, deterministic algorithms may still yield different results if subject to the stochasticity of the environment.
>
> **Focus on environmental covariate shifts**
>
> A focus of the paper, and preceding lines of work, have been in zero-shot coordination. Vanilla OBL can suffer from covariate shift during training time. Our goal was to demonstrate OBL suffers in ZSC from this covariate shift and OT-BL reduces the effects of the covariate shift. In the paper we also focus on test-time covariate shift in the ad-hoc teamplay setting, where we play with radically different teammates. We demonstrate that OT-RL can lessen test time covariate shift.
>
> **Only small number, $o=3200$ of trajectories**
>
> Thank you for pointing this out, we meant to express we used $3200$ full trajectories.

---

> > ### Author Response · Authors · 2022-08-08
> > **Following up with Reviewer nA7t**
> >
> > We again thank you for taking the time to review. We are following up to see if the reviewer had any more questions/feedback. Additionally, as a summary of our initial response:
> >
> > 1. We provide clarification on the accuracy of the belief model when jointly training the belief model and policy.
> > 2. If the equilibrium shift is larger, it will lead to a larger covariate shift. OT-RL overall broadens the training distribution and allows better performance with the target policy.
> > 3. We clarified the goals of ZSC and discuss the ability for a deterministic policy to solve the ZSC problem.
> > 4. The paper focuses on reducing train time covariate shift in the ZSC setting with OT-BL and aims to reduce test time covariate shift with OT-RL. With OT-RL the test time covariate shift also includes radically different teammates.
> >
> > Please let us know if you have any more questions.
> >
> > Best,
> > Authors of Paper12658

---

> > ### Comment · Reviewer_nA7t · 2022-08-08
> > **Response**
> >
> > Thank you for taking the time to address my raised questions.
> >
> > Replying to your points:
> > 1. One such example is the area of Bayesian Policy Reuse, of which the first search result is: "A Deep Bayesian Policy Reuse Approach Against Non-Stationary Agents" by Zheng et al.. NeurIPS'18.
> > 2. Thank you for the additional details regarding the training. I would encourage the authors to provide these details in the supplementary material of the work.
> > 3. Regarding diverse equilibria, I don't follow how this claim can be made so strongly: "OT-RL helps broaden the training distribution of the behavior policy".
> > 4. I don't believe this formulation of ZSC makes sense. If they can coordinate on a learning rule, they can coordinate on a joint policy implicitly, as the learning rule can solve the equilibrium selection problem.
> >
> > Given the author's response and the my fellow reviewer's discussion I am choosing to keep my score fixed.

---

> > > ### Author Response · Authors · 2022-08-09
> > > **Additional details**
> > >
> > > Thank you for your follow-up questions and comments. Please find our additional comments below.
> > >
> > > 1.
> > > Thank you for sharing this work. To our understanding, the Policy Reuse Problem (Def 1 in Rosman et al., 2016) is about maintaining a belief over tasks (in Zheng et al. above, maintaining a belief over partner policies in multiagent settings) and selecting the best policy for the task at hand from a pre-computed policy library $\Pi$. The belief is used at test time to inform the nature of the task (resp. partner policy) and the selection of which policy to follow to maximize expected utility.
> > >
> > > In our work, we do not have a belief over tasks or policies. We instead maintain beliefs over current environment states given past observations and a particular policy, making it an entirely different object. Our beliefs are also not used at test time, but rather during training, through the mechanism of off-belief learning. Our work could be applied to the Policy Reuse Problem, for instance by using OT-RL to populate the policy library $\Pi$. Ultimately, however, our work addresses an entirely different problem, namely the training of policies and beliefs off-team.
> > >
> > > 2.
> > > Thank you for your suggestion. We will be sure to include those details.
> > >
> > > 3.
> > > OT-RL essentially allows us to utilize one or more policies $\pi_b$ to gather experience on which we train a policy $\pi$. If the different policies have non-identical distributions over possible trajectories, then the overall distribution of experiences added to the replay buffer and used for training is broader.
> > >
> > > 4.
> > > > "If they can coordinate on a learning rule, they can coordinate on a joint policy implicitly"
> > >
> > > ZSC allows agents to agree on a high level, task-agnostic learning rule. However, it forbids them from agreeing on specific implementation details (e.g. seeds or hyperparameters) since those depend on the specific task at hand, which is only revealed after the agents are isolated.
> > >
> > > Consider the lever game in Hu et al., 2020. If agents were to agree on self-play training as a high level learning rule, then they fail to coordinate on a joint policy since self-play does not preclude symmetry breaking. Furthermore, agreeing on specific rules (e.g. "Always pick the top lever") is impossible since the Dec-POMDP is only revealed after the learning rule was chosen (e.g., the Dec-POMDP could be something entirely different, not containing any lever). In contrast, if the agents were to agree on Other-Play as a learning rule, then they would succeed in coordinating on an equilibrium in the lever game. This shows that coordinating on a task-agnostic learning rule is not sufficient to coordinate on a joint policy once the task is revealed. In summary, ZSC as discussed is challenging because it fundamentally prohibits coordinating via arbitrary symmetry breaking.
> > >
> > > We hope this answers the reviewer's questions/comments, and that they will consider updating the score if they see fit.
> > >
> > > Rosman et al., "Bayesian Policy Reuse", 2016. (https://arxiv.org/pdf/1505.00284.pdf)
> > >
> > > Zheng et al., "A Deep Bayesian Policy Reuse Approach Against Non-Stationary Agents", 2018
> > >
> > > Hu et al. “Other-Play for Zero-Shot Coordination", 2020. (https://arxiv.org/pdf/2003.02979.pdf)

---

### Official Review · Reviewer_RFqN · 2022-07-02

**Rating:** 7
**Confidence:** 3
**Soundness:** 3 good
**Presentation:** 3 good
**Contribution:** 3 good

**Summary:**

The paper introduces the concept of off-team learning, where a target joint policy is learned using a behavior joint policy to address covariate shift in off-belief learning (OBL). First off-team belief learning (OT-BL) is proposed to address covariate shift, where the belief model is iteratively updated on fictitious transitions. Second, off-team reinforcement learning (OT-RL) is proposed to address test time covariate shift with respect to unseen agents, where Q-values can be learned with belief models of any distribution. Both approaches are evaluated in the Hanabi challenge. The belief model of OT-BL is shown to produce more valid fictitious samples than OBL, while OT-RL is shown to outperform OBL in various settings.

**Questions:**

None

**Limitations:**

Limitations and potential negative societal impact are adequately addressed in the paper.

**Strengths And Weaknesses:**

- **Originality**: The idea and concepts presented in the paper are novel, interesting, and highly relevant to current applications and challenges of multi-agent systems.
- **Quality**: The paper is technically sound. The explanations and experiments are presented well and the results clearly support the claims.
- **Clarity**: The paper is well-written, clearly structured, and easy to follow.
- **Significance**: The experiment design and results indicate a clear improvement over OBL.

The authors did a great job on motivating and presenting their approach. Since a fundamental issue is addressed, a smaller toy problem like the cooperative communication game from the OBL paper would have been helpful.

Minor comments:
- The tables should highlight the best performing result in boldface for better readability.
- Text and labels in Fig. 1 and 2 are very small and hard to read when printed.

---

> ### Author Response · Authors · 2022-08-02
> **Reply to Reviewer RFqN**
>
> We thank the reviewer for their positive feedback and constructive comments. We address the comments below:
>
> **Toy game**
>
> We agree that a toy game would be helpful to motivate the problem and will try to think of an appropriate setting. We’d like to also emphasize that in the large complex setting of Hanabi, we’ve demonstrated the benefits of off-team learning.
>
> **Minor comments**
>
> We appreciate the feedback and have made adjustments to the updated submission accordingly.

---

> > ### Comment · Reviewer_RFqN · 2022-08-05
> > **Response to Authors**
> >
> > Thank you for your response. The purpose of a toy game is not necessarily to demonstrate superior performance but to show that the concepts are generally valid in a tractable/understandable way.

---

### Official Review · Reviewer_s5QM · 2022-07-11

**Rating:** 5
**Confidence:** 4
**Soundness:** 3 good
**Presentation:** 2 fair
**Contribution:** 2 fair

**Summary:**

This paper identifies weaknesses in Off-belief Learning [ICML 2021] that lead to large distribution shifts at test-time evaluation in zero-shot coordination (particularly when evaluated with human players.) The paper contributes two extensions (off-team belief learning and off-team reinforcement learning) which together improve cross-play evaluation and zero-shot coordination with another agent (rank bot) and human proxy (clone bot).

**Questions:**

During the discussion period, I'm particularly keen to hear from the authors on the following prioritized list of questions that could improve my current rating of this paper:

1) What additional evidence can the authors provide that clone-bot is a good human proxy?

2) Assuming clone-bot is a good human proxy, what evidence is there that the small improvement in proxy human-AI coordination (line 327) is a significant advancement?

3) Please clarify the issues (a)-(e) regarding specific details raised in the section on strengths and weaknesses above.

4) Please justify the very strong claim in the abstract that "Zero-shot coordination is a ... pre-requisite for human-AI coordination".

**Limitations:**

The paper includes a sufficient broader impact section. However, the closing sentence regarding "working in purely cooperative settings" is a short sighted conclusion. I encourage the authors to consider the issues raised in Section 5 of "Open Problems in Cooperative AI" Dafoe et al. Arxiv 2020.

**Strengths And Weaknesses:**

The proposed methods are original extensions to an increasingly specific approach to zero-shot coordination in Hanabi. However, the diminishing returns in improvements to this system and limited scope of its evaluation weaken the significance of this finding. Most notably the improvement in ad-hoc teamplay evaluation with rank-bot (line 326) is insignificant when considering the variance of both methods evaluated and the effect size of the improvement in proxy human-AI coordination (line 327) is tiny. Furthermore, no evidence is given in this paper that the clone bot used for proxy human-AI coordination is a good human proxy.

The overall clarity of the contribution made is significantly weakened by the paper's over reliance on references to "Off-belief Learning" [Hu et al. ICML 2021] causing the paper to not be a standalone reference to reproduce the work. For example (but not limited to) the zero-shot coordination task or cross-play metric used throughout the paper are not formally defined. For details on the critical process of training the policy and belief model readers are deferred in line 209 to "follow practices in the original OBL paper [Hu et al. ICML 2021]" and the core contributions of this paper (OT-BL and OT-RL) are then defined just by how they differ to this past work.

The specific details of how the proposed extensions differ from the past work are also incomplete and potentially incorrect in parts:

a) In Figure 1 left should the arrows into \pi from \tau_t and \tau_t+1 be reversed or the same as OT-BL (Figure 1 Right)?

b) In Figure 1 Right, the blue fictitious transitions use \pi in the illustration and \pi_L in the caption? Algorithm 1 also uses \pi and not \pi_L. Is it just the caption on Figure 1 that is incorrect?

c) The loss in Equation 1 is not defined.

d) Is it intentional that the only difference in Figure 2 left (OBL) and Figure 2 right (OT-RL) is the policy used to sample real actions?

e) On line 273, does P_{pi_L} mean both players play using policy \pi_L?

---

> ### Author Response · Authors · 2022-08-02
> **Reply to Reviewer s5QM**
>
> We thank the reviewer for their constructive comments and feedback. We address comments below:
>
> **Limited improvement in OT-RL in ad-hoc teamplay**
>
> In the ad-hoc teamplay setting, the player is unknown and can be very far from the training distribution (e.g. with rank bot or clone bot). We only argue that we can apply OT-RL to improve scores with these unseen agents. Although the improvement provided by OT-RL is limited, it is clearly beneficial to broaden the training distribution. We also achieve this result with a relatively small number (3200) of trajectories.
>
> Another important datapoint is the number of “bombouts” by each agent. “Bombing out” in Hanabi is a failure case where an invalid card is played on 3 separate occasions and the team loses all their points. In self-play, clone bot achieves a score of $19.93 \pm 0.09$ (as noted in table 4 of the OBL paper); this is in great part due to a high number of bombouts. We present in the tables below the number and distribution of bombouts per agent when Vanilla OBL/OT-RL is paired with either Clone Bot or Rank Bot (summed over all 5 seeds, 25000 games in total):
>
> | Bot | # Bombouts by OBL | # Bombouts by Rank Bot | % bombouts by OBL |
> | --- | :------------------------: | :------------------------------: | :------------------------: |
> | **Vanilla OBL** | 1619 | 5272| 23.49%|
> | **OT-RL w/ rank** | 1208 | 4840 | 19.97%|
>
>
> | Bot | # Bombouts by OBL | # Bombouts by Clone Bot | % bombouts by OBL |
> | --- | :------------------------: | :------------------------------: | :------------------------: |
> | **Vanilla OBL** | 663 | 5481 | 10.79% |
> | **OT-RL w/ clone** | 473 | 5239 | 8.28%|
>
> Notice, in this important failure case, by broadening the training distribution with OT-RL we reduce the number of bombouts caused by OBL by 25.38% with rank bot and 28.66% with  clone bot. Training with OT-RL also reduces the number of bombouts by our partner. Therefore, with unseen partners far from the training distribution, we can utilize OT-RL to broaden the training distribution and reduce the failure cases of the team.
>
> **Human-Likeness of clone bot**
>
> We don’t have studies with humans interacting with clone bot, but prior works published in highly refereed conferences have used clone bot as a proxy for human evaluation [Cui et al. 2021, Hu et al. 2021].
>
> **Addressing points a-e**
>
> a) Yes, thank you for pointing out the arrow directions. We have uploaded a new version fixing figure 1.
>
> b) That is correct, the caption should use $\pi$ instead of $\pi_\ell$.
>
> c) That is a good observation. We use the supervised learning loss function $\mathcal{L}(h|\tau^i_t) = -\sum_{k=1}^{n} \log p(h_k | \tau^i_t, h_{1:k-1}),$ as mentioned in the equation between lines 219, 220. We will update the paper to make the definition in Equation 1 complete when we introduce it.
>
> d) Yes, that is intentional. We have updated the draft emphasizing this important detail. OT-RL allows us to see broader real distributions, but only trains on fictitious transitions.
>
> e) Yes, $P_{\pi_\ell}$ means that both players are using policy $\pi_\ell$
>
> **Addressing clarity and reliance on references to the original off-belief learning paper**
>
> Addressing the reviewers concerns surrounding clarity, “Zero-shot coordination task or cross-play metric used throughout the paper are not formally defined.” We’d like to note that in line 122 we state “Under ZSC the goal is to maximise the cross-play (XP) score, the average score between independently trained agents under the same high level algorithm.” We further state in line 223 of the original submission “The XP score is the average score of all teams formed by independently trained agents.” This is a formal and straightforward definition of ZSC and XP.
>
> Also, we include training details of the policy and belief training in the originally submitted appendix as noted in the main text. Please let us know if we can provide more clarity on the above or if you feel there are other parts that are unclear.
>
> **Justification of “Zero-shot coordination is a … prerequisite for human-AI coordination”**
>
> We agree the statement is strong. We know there are methods that can rely on behavior cloning, where human data is sufficient to achieve human-AI coordination including but not limited to Carroll et al. 2019. We’ve toned down the language in the abstract.

---

> > ### Comment · Reviewer_s5QM · 2022-08-03
> > **Outstanding Concerns**
> >
> > Thank you for the detailed reply. I appreciate your responses to points a-e, addressing the reliance on references to the original off-belief learning paper and the justification of zero-shot coordination as a prerequisite for human-AI coordination. Incorporating these responses into the revised paper will improve its overall presentation and I will update my scores to reflect this change in my position at the end of the discussion period.
> >
> > However, I remain in doubt about the significance of the contribution so hope to continue the discussion regarding the first two questions I raised.
> >
> > 1. Regarding human-likeness of clone bot. I did not intend to ask for "studies with humans interacting with clone bot." Instead, I am seeking evidence that clone bot's policy is a good model of human policies. This would strengthen the argument that the difference between OBL and OT-RL when playing with clone bot might also happen when playing with human players, and is not just overfitting to clone bot.
> >
> > 2. Regarding the limited improvement in proxy human-AI coordination with clone bot. Thank you for this additional analysis and insight into the frequency of specific failure cases. However, whilst the effect size here is larger (due to the range of the statistic measured), reducing the number of "bombouts" is not the goal of ZSC as defined by the authors in their own reply. Therefore, I would argue that the need to move to this second level of analysis is further evidence of the diminishing returns in improvements to this system.
> >
> > Together, these issues lead me to currently conclude that the contribution is incremental as it is only evaluated empirically in one application where the effect shown is small and the correlation between this effect and performance on the true goal of human-AI collaboration is unknown. I remain open to changing this conclusion and look forward to discussing this further with the authors and other reviewers.

---

> > > ### Author Response · Authors · 2022-08-05
> > > **Reply to Reviewer s5QM**
> > >
> > > We again thank you for taking the time to review. We address comments below:
> > >
> > > **Human-likeness of clone bot**
> > >
> > > In Jacob et al. 2022, they run an experiment where they take the same dataset used to train clone-bot, 240,954 2-player Hanabi games and randomly sample 4,000 games for a test set. They train a new clone-bot using the same supervised learning procedure on the smaller training set. We summarize the human-move prediction below:
> > >
> > > | Subset of Test Set | % of Moves Matched |
> > > | --- | :------------------------: |
> > > | All Games | 63.63% ± 0.18%  |
> > > | Games w/ score ≥ 10 | 65.29% ± 0.19% |
> > > | Games w/ score ≥ 20 | 67.39% ± 0.19% |
> > >
> > > Notice, that the percentage of moves matched across all hold out test games for this clone bot is > 63%. Considering the dataset was composed of players of all levels and that humans can disagree on moves to make, this indicates that clone bot is reasonably human-like. Let us know if you have any questions or would like more clarification.
> > >
> > > **Bombouts in ZSC**
> > >
> > > First, we’d like to emphasize that ZSC, as specified in our reply and formalized in Hu et al. 2020, is measured through the XP of agents independently trained using the same algorithm. Our ZSC results are presented in Tables 1 and 2, where OT-BL and OT-RL significantly outperform Vanilla OBL in both 2 and 3-life Hanabi.
> > >
> > > Secondly, In Ad-hoc coordination (Table 3), the reviewer is correct in saying that the improvement brought by OT-RL is small (scores are improved, but the difference is less than the sum of standard errors on the mean). While we agree that avoiding bombouts is not an explicit goal, they are still important. This is because bombouts are a telltale sign of covariate shifts. They occur when a player attempts to play a non-playable card when the team has only one life left, losing all accumulated points and ending the game prematurely. In Hanabi, an agent always has alternatives to playing a card (i.e. discarding or hinting), so bombouts only happen if an agent either misunderstood the hints given by its partner or if its Q-value predictions are inaccurate, both of which can be caused by covariate shifts.
> > >
> > > Our analysis shows that OT-RL significantly reduces the instances where our agent causes such bombouts, plus an additional small positive effect on the bombout rate of our partner, which itself is also subject to covariate shift when paired with OT-RL.
> > >
> > >
> > > **References**
> > >
> > > [1] H. Hu, A. Lerer, A. Peysakhovich, and J. Foerster. “Other-play” for zero-shot coordination. In 387 H. D. III and A. Singh, editors, Proceedings of the 37th International Conference on Machine 388 Learning, volume 119 of Proceedings of Machine Learning Research, pages 4399–4410. PMLR, 389 13–18 Jul 2020.
> > >
> > > [2] A. Jacob, D. Wu, G. Farina, A. Lerer, H. Hu, A. Bakhtin, J. Andreas, & N. Brown. (2022). Modeling Strong and Human-Like Gameplay with KL-Regularized Search. Proceedings of the 39th International Conference on Machine Learning. Proceedings of Machine Learning Research. Jul 2022.

---

> > > > ### Comment · Reviewer_s5QM · 2022-08-08
> > > > **Bombouts + Human-Likeness**
> > > >
> > > > Ok, I now see the value in measuring bombouts to demonstrate OT-RL reduces covariate shift in Hanabi. This improves my view on the significance of the contribution. I have updated the scores in my original review to reflect this assuming the additional analysis of bombouts will be included in the camera-ready version of the paper if accepted.
> > > >
> > > > I also appreciate the additional data on the accuracy of clone bot. However, there is clearly an open opportunity to improve the human-likeness of clone bot (as demonstrated by pikl-SPARTA [Jacob et al. 2022]). I would also still argue that the correlation in performance between collaborating with clone bot and humans is unknown. Improving this aspect of the evaluation is obviously out of scope for consideration within the NeurIPS review period, but an interesting direction for future work.

---

> > > ### Author Response · Authors · 2022-08-08
> > > **Following up with Reviewer s5QM**
> > >
> > > We again thank you for taking the time to review. We are following up to see if the reviewer had any more questions/feedback. Additionally, as a summary of our last response:
> > >
> > > 1. Jacob et al. 2022 trained a new clone-bot on a smaller set of training data. On a holdout set of games, the bot matched the moves > 63% of the time. Given the entire dataset is comprised of players of various levels and humans can even disagree on moves, clone-bot is reasonably human-like.
> > > 2. Although bombouts are not an explicit goal, they are a telltale sign of covariate shifts. With OT-RL and a limited number of target trajectories, we are able to significantly reduce the number of bombouts by our trained agents.
> > >
> > > Please let us know if you have any more questions.
> > >
> > > Best,
> > > Authors of Paper12658

---

### Meta-Review · Area_Chair_dnJp · 2022-08-26

**Recommendation:** Accept
**Confidence:** Less certain

**Metareview:**

The authors propose an improvement to off-belief learning, Off-Team learning, which closes the gap between belief models trained on fixed policies, and evaluation on learned policies for ZSC coordination problems. All reviewers have voted to weak/borderline accept - since I see no conceptual issues with the proposed framework and the evaluation seems sound, I will also vote to accept. The major area of constructive criticism, is that the work seems to be somewhat incremental with respect to Hu et al. (ICML 21).


**Award:**

No

---

### Decision · Program_Chairs · 2022-09-14

Accept